# G-Refer: Graph Retrieval-Augmented Large Language Model for Explainable Recommendation

## ABSTRACT

Explainable recommendation has demonstrated significant advantages in informing users about the logic behind recommendations, thereby increasing system transparency, effectiveness, and trustworthiness. To provide personalized and interpretable explanations, existing works often combine the generation capabilities of large language models (LLMs) with collaborative filtering (CF) information. CF information extracted from the user-item interaction graph captures the user behaviors and preferences, which is crucial for providing informative explanations. However, due to the complexity of graph structure, effectively extracting the CF information from graphs still remains a challenge. Moreover, existing methods often struggle with the integration of extracted CF information with LLMs due to its implicit representation and the modality gap between graph structures and natural language explanations. To address these challenges, we propose **G-Refer**, a framework using Graph Retrieval-augmented large language models (LLMs) for explainable recommendation. Specifically, we first employ a hybrid graph retrieval mechanism to retrieve explicit CF signals from both structural and semantic perspectives. The retrieved CF information is explicitly formulated as human-understandable text by the proposed graph translation and accounts for the explanations generated by LLMs. To bridge the modality gap, we introduce knowledge pruning and retrieval-augmented fine-tuning to enhance the ability of LLMs to process and utilize the retrieved CF information to generate explanations. Extensive experiments show that G-Refer achieves superior performance compared with existing methods in both explainability and stability. Codes and data are available at https://anonymous.4open.science/r/G-Refer.

## CCS CONCEPTS

• **Computing methodologies → Artificial intelligence**; • **Information systems → Information retrieval**.

## KEYWORDS

Explainable Recommendation, Large Language Model, GraphRAG

**ACM Reference Format:**
Anonymous Author(s). 2018. G-Refer: Graph Retrieval-Augmented Large Language Model for Explainable Recommendation. In *Proceedings of (Conference acronym 'XX)*. ACM, New York, NY, USA, 12 pages. https://doi.org/10.1145/nnnnnnn.nnnnnnn

## 1 INTRODUCTION

Recommendation systems (RS) play an essential role in our daily lives, as they affect how people navigate the vast amounts of products available in online services [21, 37, 56]. These systems have the ability to predict whether a user is interested in an item, *e.g.*, by clicking on it or making a purchase. Recently, explainable recommendation [45, 55, 70, 71] has attracted more attention and demonstrated significant advantages in informing users about the logic behind their received recommendation results, thereby increasing system transparency, effectiveness, and trustworthiness. Specifically, explainable recommendation aims to generate human-understandable textual explanations for each user-item recommendation. This enables us to understand the underlying reasons behind each recommendation and develop accountable RS as a result [7].

Due to the impressive generative and reasoning capabilities of large language models (LLMs) [74], existing explainable recommendation methods [30, 31, 39] can now produce fluent and informative explanations in natural language based on user and item profiles, as well as their interactions. To provide more detailed and personalized explanations, recent works [39] have utilized user-item interaction graphs, which contain abundant collaborative filtering (CF) information from graph structure. The CF information [22] reveals the complex patterns between users and items, which can be used to generate more accurate and informative explanations. Nevertheless, the graph structure is inherently noisy and complex, making it challenging to effectively extract CF information for explanations.

In order to better model the CF information from graph structure in RS, graph neural networks (GNNs) have been widely adopted and demonstrated exceptional performance in recommendation tasks [2, 20, 57]. GNNs capture CF information by learning hidden representations of both users and items through an iterative process of feature transformation and information aggregation from neighboring nodes. While GNNs excel at capturing CF information for recommendations, simply incorporating GNNs with LLM-based RS such as XRec [39] still faces several challenges in producing satisfactory explanations for recommendations: **(C1) Implicit CF signal.** The CF signals are injected into LLMs by feeding user and item embeddings generated by GNNs, where the CF signals are represented as implicit node embeddings. Given the vagueness of the explainability of GNNs themselves [65], it is difficult to interpret the CF signals contained in these embeddings. Therefore, more explicit evidence constructed from the CF signals contained in graphs, especially favored from the view of human-understandable text, is eagerly needed. **(C2) Modality gap.** Since GNNs are primarily used to capture structural information, the rich semantics within user and item profiles are inevitably ignored. In addition, LLMs struggle to directly understand the structured CF signals captured by GNNs, because of their unstructured nature. This results in a significant modality gap between structured representations and

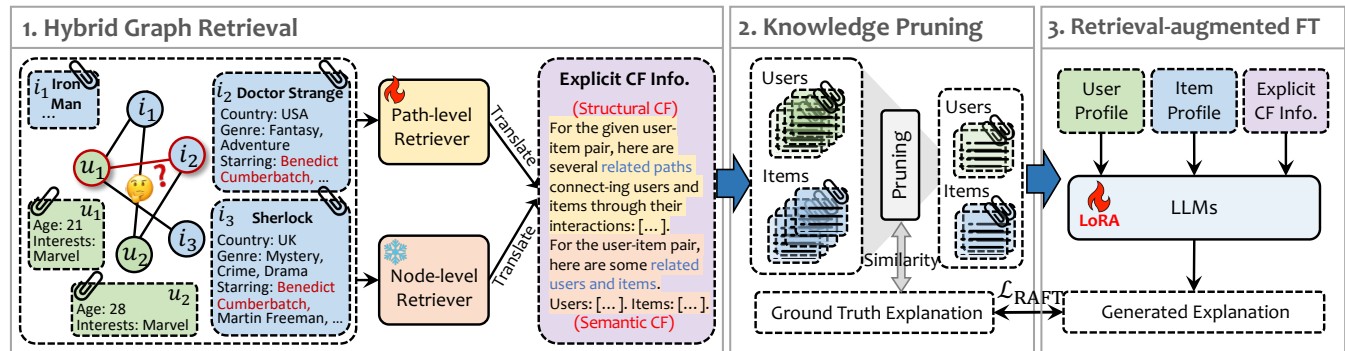

**Figure 1: Our proposed pipeline G-Refer facilitates explainable recommendation with three key components: (1) Hybrid Graph Retrieval employs multi-granularity retrievers to retrieve explicit CF signals and formulated as human-readable text by the Graph Translation; (2) Knowledge Pruning eliminates noise and improves training efficiency; and (3) Retrieval-augmented Fine-tuning instructs LLMs to leverage retrieved CF information in generating informative explanation.**

natural language, making it difficult for LLMs to effectively leverage graph-derived CF information when generating explanations.

**Presented Work.** Motivated by these challenges, in this work, we propose G-Refer, a framework designed to use Graph Retrieval-augmented LLMs for explainable recommendation. We aim to address the limitations of existing methods and extract explicit, diverse, and semantically rich CF information to generate more accurate and personalized explanations. As shown in Figure 1, given a user-item recommendation to be explained, we leverage a hybrid graph retrieval mechanism that combines multi-granularity retrievers to extract explicit *structural* and *semantic* CF information from the graphs (to address **C1**). Specifically, we employ a path-level retriever to capture structural CF information by identifying $k$ paths from graphs that account for recommendation. To utilize the rich semantics within nodes, we also employ a node-level retriever to retrieve semantic CF information from the graphs by finding the most relevant nodes to the recommendation. The retrieved CF information is then translated into human-understandable text with the graph translation module to facilitate LLMs in generating explanations. To bridge the modality gap and enhance the understanding of LLMs (to address **C2**), we adopt a lightweight retrieval-augmented fine-tuning (RAFT) approach to instruct LLMs in understanding the retrieved CF information and generating explanations. To further improve the training efficiency and reduce noise, we introduce a knowledge pruning technique to filter out training samples with less relevant CF information. After the training, LLMs exhibit a greater ability to leverage both the retrieved CF information and rich semantics in profiles to generate accurate and contextually relevant explanations for recommendations.

Our main contributions can be summarized as follows:

- **Comprehensive Analysis.** We identify the challenges in existing explainable recommendation works, as GNNs struggle to capture explicit and semantically rich CF information.
- **Architecture Design.** We propose G-Refer, a model leveraging hybrid graph retrieval to capture both structural and semantic CF signals from user-item interaction graphs. We also incorporate knowledge pruning and retrieval-augmented fine-tuning to enhance LLMs' ability to utilize retrieved knowledge.

- **Superior Performance.** Extensive experiments on public datasets demonstrate the effectiveness of our proposed G-Refer, surpassing a series of SOTA baselines by up to 8.67%.

## 2 PRELIMINARIES

### 2.1 Explainable Recommendation

The explainable recommendation aims to unveil the rationale behind the recommendation results and provide human-understandable textual explanations. In recommendation scenarios, the user-item interactions can be represented by a bipartite graph $\mathcal{G} = \{(u, i)|u \in \mathcal{U}, i \in \mathcal{I}\}$, where $\mathcal{U}$ and $\mathcal{I}$ are the sets of users and items, respectively. Each node in $\mathcal{G}$ is associated with textual information from either the user profile set $\mathcal{B}$ or the item profile set $C$. Specifically, each user $u$ has a profile $b_u \in \mathcal{B}$ describing their preferences, and each item $i$ has a profile $c_i \in C$ detailing attributes and descriptions.

Given a recommendation $(u, i)$, which is a pair of user $u$ and recommended item $i$, the goal of an explainable recommendation is to generate a clear textual explanation, which is formulated as:

$$explanation(u, i) = \text{generator}(u, i, \mathcal{G}, \mathcal{B}, C). \quad (1)$$

The graph models abundant structural CF information beneficial for explainable recommendations. Given the size of the entire graph, we commonly utilize the subgraph $\mathcal{G}(u, i) \subseteq \mathcal{G}$ as input [63, 67]. This subgraph represents the $L$-hop edge-centered ego-graph around the pair $(u, i)$. It includes nodes that are at most $L$ hops away from either $u$ or $i$, effectively capturing the local connectivity structure for generating explanations.

### 2.2 Graph Retrieval-Augmented Generation

Retrieval-Augmented Generation (RAG) has achieved remarkable success in enhancing Large Language Models (LLMs) by retrieving external documents. However, existing RAGs fail to fully exploit the structural information in the graph to augment LLMs. The graph retrieval-augmented generation (GraphRAG), to address this issue, leverages structural information to enable more precise retrieval and facilitate context-aware generation [46]. Given a query $q$, GraphRAG aims to retrieve structural information from a graph $\mathcal{G}$ and generate responses $a$ conditioned by the retrieved information,

which can be formulated as:

$$G = \text{G-retriever}(q, \mathcal{G}), \tag{2}$$

$$P(a|q, \mathcal{G}) = P(a|q, G, \theta), \tag{3}$$

where $\theta$ is the parameters of the LLMs, and $G$ is the structural information (e.g., nodes, paths, and subgraphs) retrieved by the graph retriever.

## 3 METHODOLOGY

Our proposed G-Refer is designed to excavate the CF information from the graph to enhance explainable recommendations. First, we leverage a hybrid graph retrieval mechanism that combines multi-granularity retrievers to extract both structural and semantic CF information from the user-item interaction graph. In the second part, we employ knowledge pruning to filter out less relevant or redundant training samples, eliminating noise and improving training efficiency. Finally, a lightweight retrieval-augmented fine-tuning is conducted to bridge the modality gap, instructing LLMs to understand retrieved knowledge and generate accurate and contextually relevant explanations.

### 3.1 Hybrid Graph Retrieval

The user-item interaction graph encapsulates a wealth of knowledge, including user behaviors, preferences, and relationships among users and between items [56]. It can provide sufficient CF information from different perspectives, such as **structural** and **semantic** views, to explain the user behaviors, thereby enhancing the explainability of recommendations. For example, as illustrated in Figure 1, we can explain why user $u_1$ might be interested in item $i_2$ from both structural and semantic perspectives. From the structural perspective, the path ($u_1 \rightarrow i_1 \rightarrow u_2 \rightarrow i_2$) illustrates the potentially shared preferences between user $u_1$ and $u_2$, both of whom have watched $i_1$, *Iron Man*, suggests that user $u_1$ may also enjoy $i_2$, *Doctor Strange*, which user $u_2$ has watched, due to the overlapping Marvel fandom. In addition, from the semantic view, retrieving items $i_2$ and $i_3$ could also lead to an explanation. Both items feature the same leading actor, Benedict Cumberbatch. This commonality suggests that user $u_1$, having shown interest in related actors through their viewing habits, might find $i_3$, *Sherlock*, appealing as well.

According to [46], retrievers based on different levels of granularity in the graph (e.g., nodes, paths, and subgraphs) have unique strengths for addressing various aspects of retrieval scenarios. Inspired by recent works [16, 38], we adopt a hybrid graph retrieval strategy that employs both **path-level** and **node-level** retrievers to respectively retrieve structural and semantic CF information from the user-item interaction graph, which is composed of purchase histories with enriched text attributes. The CF information retrieved from both views is then translated into human-understandable text, facilitating the generation of explanations.

*3.1.1 Path-level Retriever.* The path-level retriever aims to retrieve the structural CF information from the graphs by identifying the most $k$ informative paths that connect users and items for interpretable recommendations. Typically, recommendations on graphs can be formalized as a link prediction (LP) problem, which predicts the connections between user $u$ and $i$ based on the graph structure

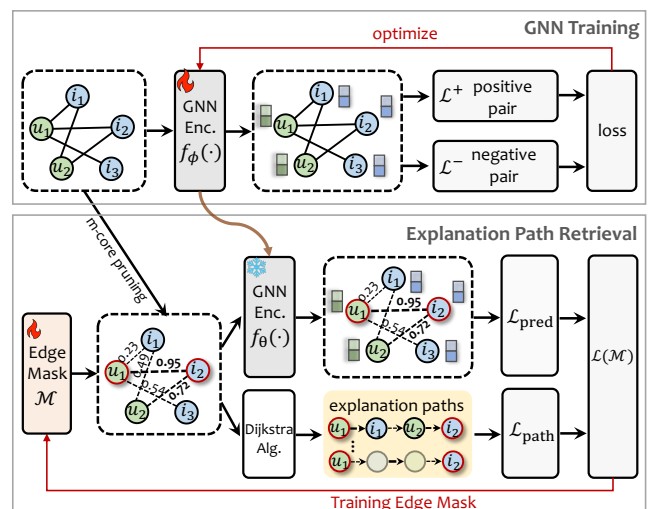

**Figure 2: The illustration of path-level retriever.**

[33]. This can be formulated as

$$i = f_\phi(u, \mathcal{G}), \tag{4}$$

where $f_\phi$ denotes a link prediction model parameterized by $\phi$. The $f_\phi$ captures complex patterns between users and items in the graph to provide the recommendation results.

**Motivation.** GNNs [58] have demonstrated exceptional performance in capturing graph structures and are widely used in recommender systems [20, 57]. While GNNs excel at extracting CF information for recommendations, they are often criticized for their lack of interpretability. GNNs struggle to provide clear explanations for their predictions [64], which significantly undermines transparency. To address this issue, inspired by recent GNN explanation methods [67], we propose a path-level retriever that generates explanations for GNN predictions and reveals the structural CF information captured by GNNs as interpretable paths, which is illustrated in Figure 2.

**GNN Training.** To capture the structural CF information, we first train a GNN to learn the user-item interactions in the graph for recommendations. We follow a standard two-step pipeline, which involves obtaining user and item representations with an $L$-hop GNN encoder and applying a prediction head to get a recommendation prediction. We use R-GCN [49] as the GNN encoder to learn node embeddings on user-item graphs, and an inner product is adopted as the prediction head. We also explore other GNN variants, i.e., LightGCN [20], in our experiments (Section 4.3). The forward-pass update of each user $u$ in R-GCN is formalized as follows:

$$\mathbf{h}_u^{(l)} = \text{ReLU}\left(\sum_{i' \in \mathcal{N}_u} \frac{1}{|\mathcal{N}_u|} \mathbf{W}_1^{(l-1)} \mathbf{h}_{i'}^{(l-1)} + \mathbf{W}_0^{(l-1)} \mathbf{h}_u^{(l-1)}\right), \tag{5}$$

where $\mathbf{h}_u^{(l)}$ is the hidden state of user $u$ in the $l$-th layer and $\mathcal{N}_u$ denotes the neighbors of $u$. The $\mathbf{W}_0^{(l-1)}$ and $\mathbf{W}_1^{(l-1)}$ are learnable weight matrices at layer $l$-1. Likewise, the item representation $\mathbf{h}_i^{(l)}$ can be obtained in a similar manner.

**The $m$-core Pruning.** To ensure concise and accurate explanations, we aim to extract paths that are short in length and avoid containing

high-degree nodes in the graph [67]. Given a prediction $(u, i)$, the corresponding $L$-hop edge-centered ego-graph is initially extracted, which includes nodes that are at most $L$ hops away from either $u$ or $i$. A $m$-core pruning process is then applied to remove spurious neighbors and improve speed. Specifically, the $m$-core pruning is a recursive algorithm that eliminates nodes with degrees less than $m$, until only those with degrees $\geq m$ remain, forming the $m$-core.

**Explanation Path Retrieval.** We follow PaGE-Link [67] to perform mask learning on the learned GNN model to identify explanation paths from the user-item interaction graph via learning a mask over all edges, assigning high weights to important edges and low weights to others. Based on the mask, we apply the Dijkstra's shortest path algorithm [13] to retrieve explanation paths.

Formally, let $\mathcal{E}$ denote all user-item interactions, $\tau(e)$ be the type of the edge $e$, and $\mathcal{M}$ be learnable masks of all edge types. To optimize $\mathcal{M}$, the objective function $\mathcal{L}$ that needs to be minimized is composed of two loss terms, which can be formulated as follows:

$$\mathcal{L}(\mathcal{M}) = \mathcal{L}_{pred}(\mathcal{M}) + \mathcal{L}_{path}(\mathcal{M}). \quad (6)$$

The first loss term $\mathcal{L}_{pred}$ is to learn to select crucial edges for model prediction based on the perturbation-based explanation. Given a connected pair $(u, i)$, the loss can be defined as follows:

$$\mathcal{L}_{pred}(\mathcal{M}) = -\log P(Y = 1|\mathcal{G} = (\mathcal{U}, \mathcal{I}, \mathcal{E} \odot \sigma(\mathcal{M})), (u, i)), \quad (7)$$

where $\sigma(\cdot)$ is the sigmoid function, $\odot$ is the element-wise product, and $Y$ is the original GNN prediction. Another loss term $\mathcal{L}_{path}$ is to learn to select path-forming edges. We first use the following score function to assign a score to each candidate path $p$ in the graph, prioritizing shorter paths and those without high-degree nodes:

$$Score(p) = \prod_{e \in p} \log \sigma(\mathcal{M}_{\tau(e)}) - \log(\deg(e)), \quad (8)$$

where $\mathcal{M}_{\tau(e)}$ denotes the mask corresponding to the type of the edge $e$ and $\deg(e)$ is the degree of the target node of $e$. Using Dijkstra's shortest path algorithm, where path lengths are redefined according to the score function, a set of candidate edges considered concise and informative is selected, denoted by $\mathcal{E}_{path}$. Guided by $\mathcal{E}_{path}$, we define $\mathcal{L}_{path}$ accordingly:

$$\mathcal{L}_{path}(\mathcal{M}) = -\sum_{r \in \mathcal{R}} \left( \sum_{e \in \mathcal{E}_{path}} \mathcal{M}_{\tau(e)} - \sum_{e \in \mathcal{E}, e \notin \mathcal{E}_{path}} \mathcal{M}_{\tau(e)} \right). \quad (9)$$

Upon convergence of mask learning, we can run the Dijkstra's shortest path to generate up to $k$ explanation paths based on $\mathcal{M}$.

*3.1.2 Node-level Retriever.* The node-level retriever aims to retrieve the semantic CF information from the graphs by selecting the $k$ most relevant user nodes and the $k$ most relevant item nodes.

**Motivation.** Graphs are often known as noisy and incomplete, and the structural information alone may not be sufficient to provide accurate explanations. Additionally, graphs in recommendations typically contain rich semantics within user and item nodes derived from user interests and item properties. While the profiles provide some information, using only individual profiles for explanations may be inadequate, as it might not fully capture user topics or item features. Thus, retrieving semantics from graphs is crucial to uncovering latent connections and supplementing more comprehensive explanations for recommendations. Our node-level retriever incorporates a dense retrieval to calculate the semantic

similarity between users and items to retrieve the most relevant neighboring nodes for the current recommendation $(u, i)$ to unveil the semantic CF knowledge for explanations.

**Dense Retrieval.** We employ a dual-encoder-based retriever architecture, which has demonstrated effectiveness across various retrieval tasks [42] and efficiency at the inference stage [24, 27]. Given a user-item pair $(u, i)$, a text encoder maps the user profile $b_u$ to an embedding $f(b_u) \in \mathbb{R}^d$, and the same encoder maps each item profile $c_i$ to an embedding $f(c_i) \in \mathbb{R}^d$, where $d$ is the hidden state dimension of the text encoder. The top-$k$ relevant users are retrieved based on the user-user semantic similarities, which are computed via cosine similarity:

$$sim(u, u') = \frac{f(b_u) \cdot f(b_{u'})}{\|f(b_u)\| \|f(b_{u'})\|}, \quad u' \in \mathcal{N}_i, \quad (10)$$

where $\mathcal{N}_i$ is the set of users connected to item $i$. We also retrieve top-$k$ items based on the item-item semantic similarities. The retriever is training-free and a pre-trained language model specifically utilized to generate high-quality sentence embeddings.

*3.1.3 Graph Translation.* After the hybrid retrieval, the complex nature of graph-type knowledge, particularly the paths retrieved, presents a challenge since it cannot be directly integrated with profiles of users and items for input into the LLMs [6, 53]. To address this, it is necessary to employ graph translation techniques that convert the graph-type knowledge into a format compatible with LLMs, enabling LLMs to effectively process and utilize structured information. Considering that instructions are typically presented in natural language, we follow [52, 62] to adopt a flatten-based method, which transforms retrieved paths and nodes into descriptive, easily comprehensible language. For each user-item pair $(u, i)$ requiring an explanation, along with $k$ retrieved paths, $k$ user nodes, and $k$ item nodes, we design the instruction prompt as follows:

---

Given the item title, item profile, and user profile, please explain why the user would enjoy this item. Item title: [Item title]. Item profile: [Item summary]. User profile: [User summary]. For the user-item pair, here are some related users and items. Users: [Top-$k$ similar user profiles]. Items: [Top-$k$ similar item profiles]. For the given user-item pair, here are several related paths connecting users and items through their interactions. [Top-$k$ explanation paths] (each formatted as "<User profile> -> buys -> <Item profile> -> bought by -> <User profile> -> ..."). Explanations:

---

*3.1.4 Discussion of Retrieval Granularity.* In G-Refer, we consider node-level and path-level retrieval rather than subgraph-level based on the following observations. Firstly, both nodes and paths have a considerably smaller search space than subgraphs. As proven in [67], compared to the expected number of edge-induced subgraphs, the expected number of candidate nodes and paths grows strictly slower and becomes negligible. Therefore, such explanations exclude many less-meaningful subgraph candidates, making the explanation generation much more straightforward and accurate. Secondly, subgraphs contain complex high-order information, making it difficult to describe a retrieved subgraph with hundreds or even thousands of nodes in a way that is easily understandable, especially when all nodes have text attributes [36]. Consequently, we implement hybrid graph retrieval in the path and node levels.

## 3.2 Knowledge Pruning

**Motivation.** After graph retrieval, our goal is to enable LLMs to effectively generate explanations based on the graph CF information of varying granularities retrieved. However, it is noticed that for some user-item pairs, a sufficient explanation can be derived solely from their profiles, without the need for additional CF information. For instance, if a user's profile shows a preference for *science fiction* and the recommended item is a popular *sci-fi film*, then explaining this interaction might only require referencing these profile details. In such cases, the retrieved knowledge is not only redundant but employing such training samples could also weaken the LLM's ability to utilize CF information.

**Re-ranking.** To enhance LLMs' focus on CF information, we propose knowledge pruning to filter out less relevant or redundant training samples [46], building on the inherent ability of LLMs to generate explanations by understanding user and item profiles. It also improves training efficiency through reduced data volume, particularly beneficial for large graphs as indicated in Table 1. To be specific, we use a re-ranking method that sorts all training samples based on their reliance on CF information for explanation, prioritizing those that require more additional knowledge. We measure such reliance via the semantic similarity between the profiles and the ground truth explanations, which is defined as follows:

$$sim((u,i), \text{Explain}_{(u,i)}) = \frac{f(b_u \oplus c_i) \cdot f(\text{Explain}_{(u,i)})}{\|f(b_u \oplus c_i)\| \|f(\text{Explain}_{(u,i)})\|}, \quad (11)$$

where $\oplus$ represents the concatenation and $\text{Explain}_{(u,i)}$ is the ground truth explanation for $(u, i)$. We introduce a pruning ratio $t$, which determines the portion of the total training data $\mathcal{D}$ to be filtered after re-ranking. The top "$(1-t) \cdot |\mathcal{D}|$" training samples are preserved, ensuring that samples that are most likely to benefit from CF-based explanations are prioritized, thereby improving the overall efficiency and effectiveness of the training process.

## 3.3 Retrieval-Augmented Fine-Tuning

**Motivation.** To improve the LLM's capacity for generating better explanations using retrieved CF information, we fine-tune it with a parameter-efficient pre-training strategy, i.e., LoRA [23], on the pruned training set with in-context retrieval augmentation. Integrating retrieval results during fine-tuning offers two key advantages. (1) It adapts the LLM to better utilize retrieved CF information to generate explanation, especially for the requirement of domain-specific knowledge it has never seen before. (2) Even state-of-the-art retrievers can falter and return inaccurate results [35]. By training the LLM to generate ground-truth responses even when irrelevant CF information is given, we enable the LLM to ignore misleading retrieval content and lean into its internal knowledge to reduce hallucination.

**Fine-tuning.** We employ retrieval-augmented fine-tuning (RAFT) with a conventional language modeling loss [69]. The model is trained to process input consisting of a user-item pair $(u, i)$ with profiles $b_u$ and $c_i$, its associated retrieved knowledge $\mathcal{K}_{(u,i)}$, and a corresponding prompt question $Q$ to generate textual explanations $E$ as outputs. The RAFT loss is computed based on the discrepancy between the model's predicted explanations and the ground truth

**Table 1: Statistics of the experimental datasets.**

| Dataset | # Users | # Items | # Interactions | # Train $(u-i)$ | # Test $(u-i)$ |
|---|---|---|---|---|---|
| **Amazon-books** | 15,349 | 15,247 | 360,839 | 95,841 | 3,000 |
| **Yelp** | 15,942 | 14,085 | 393,680 | 74,212 | 3,000 |
| **Google-reviews** | 22,582 | 16,557 | 411,840 | 94,663 | 3,000 |

explanations in the dataset. Formally, the loss is defined as follows:

$$\mathcal{L}_{\text{RAFT}} = -\sum_{(u,i) \in \mathcal{D}_{\text{prune}}} \log P(\text{Explain}_{(u,i)}|b_u, c_i, \mathcal{K}_{(u,i)}, Q; \theta), \quad (12)$$

where $\mathcal{D}_{\text{prune}}$ represents the pruned training set and $\theta$ is the parameters associated with the LoRA model.

## 4 EXPERIMENTS

We evaluate our model on real-world datasets to assess its performance in enhancing explainable recommendations. In particular, we aim to answer the following research questions: **Q1**: How effective is G-Refer compared with state-of-the-art models? **Q2**: How do the main components of our model impact the performance? **Q3**: What is the impact of hyper-parameters? **Q4**: How does G-Refer's training efficiency compare to state-of-the-art methods?

### 4.1 Experimental Protocols

*4.1.1 Datasets.* We evaluate G-Refer and leverage three prominent public datasets that offer distinct perspectives on user-item interactions, including **Amazon-books** [41], **Yelp** [39], and **Google-reviews** [28, 60]. Table 1 lists the statistics of three datasets. More details of datasets can be found in Appendix A.1.

*4.1.2 Metrics.* We follow XRec [39] to utilize a suite of metrics aimed at assessing the semantic explainability and stability of the generated explanations. Traditional n-gram-based metrics like BLEU [44] and ROUGE [34] are not adequate for this purpose due to their inability to fully capture semantic meaning. Specifically, we use **GPT**$_{\text{score}}$ [54], **BERT**$_{\text{score}}$ [68], **BART**$_{\text{score}}$ [66], **BLEURT** [50], and **USR** [30] to measure the explainability. Notably, BERT$_{\text{score}}$ comprises the Recall score, which measures the completeness and quality of the retrieved CF information, allowing us to evaluate the performance of our graph retriever. More details of the used metrics can be found in Appendix A.2. To evaluate quality consistency, we also report the standard deviations of these metrics. Lower standard deviation values indicate more stable performance.

*4.1.3 Baselines.* We introduce five state-of-the-art baselines, including **NRT** [32], **Att2Seq** [14], **PETER** [30], **PEPLER** [31], and **XRec** [39]. These models are based on representative language models, such as GRU [10], LSTM [19], GPT, and LLaMA. More details of the compared baselines can be found in Appendix A.3.

*4.1.4 Implementation Details.* For path-level retrieval, we set the node embedding dimension to 128, the maximum retrieved path length to 5, and $m$ is set to 2 for m-core pruning. We initialize the encoders in both the node-level retriever and the re-ranking mechanism with SentenceBERT [47]. The number of retrieved paths, as well as the number of retrieved nodes (including users and items), is set to 2, and the pruning ratio $t$ is set to 70% across all datasets.

**Table 2: Overall comparison in terms of Explainability and Stability. Superscripts "P", "R", and "F1" denote Precision, Recall, and F1-Score, respectively. The subscript "std" indicates the standard deviation of each metric. Bold indicates the best results, while underlined denotes the second-best. "7B" and "8B" denote LLaMA 2-7B and LLaMA 3-8B, respectively.**

| Models | Explainability ↑ | | | | | | | Stability ↓ | | | | | |
|---|---|---|---|---|---|---|---|---|---|---|---|---|---|
| | $GPT_{score}$ | $BERT_{score}^P$ | $BERT_{score}^R$ | $BERT_{score}^{F1}$ | $BART_{score}$ | BLEURT | USR | $GPT_{std}$ | $BERT_{std}^P$ | $BERT_{std}^R$ | $BERT_{std}^{F1}$ | $BART_{std}$ | $BLEURT_{std}$ |
| **Amazon-books** | | | | | | | | | | | | | |
| NRT | 75.63 | 0.3444 | 0.3440 | 0.3443 | -3.9806 | -0.4073 | 0.5413 | 12.82 | 0.1804 | 0.1035 | 0.1321 | 0.5101 | 0.3104 |
| Att2Seq | 76.08 | 0.3746 | 0.3624 | 0.3687 | -3.9440 | -0.3302 | 0.7757 | 12.56 | 0.1691 | 0.1051 | 0.1275 | 0.5080 | 0.2990 |
| PETER | 77.65 | **0.4279** | 0.3799 | 0.4043 | -3.8968 | -0.2937 | 0.8480 | 11.21 | 0.1334 | 0.1035 | 0.1098 | 0.5144 | 0.2667 |
| PEPLER | 78.77 | 0.3506 | 0.3569 | 0.3543 | -3.9142 | -0.2950 | 0.9563 | 11.38 | 0.1105 | 0.0935 | 0.0893 | 0.5064 | 0.2195 |
| XRec | 82.57 | 0.4193 | 0.4038 | 0.4122 | -3.8035 | **-0.1061** | **1.0000** | 9.60 | **0.0836** | 0.0920 | **0.0800** | 0.4832 | **0.1780** |
| G-Refer (7B) | 82.70 | 0.4076 | 0.4476 | 0.4282 | -2.4772 | -0.1246 | **1.0000** | 9.04 | 0.0937 | **0.0845** | 0.0820 | 0.2769 | 0.1893 |
| G-Refer (8B) | **82.82** | 0.4073 | **0.4494** (+4.56%) | **0.4289** (+1.67%) | **-2.4699** | -0.1203 | **1.0000** | **8.95** | 0.0945 | 0.0855 | 0.0825 | 0.2791 | 0.1912 |
| **Yelp** | | | | | | | | | | | | | |
| NRT | 61.94 | 0.0795 | 0.2225 | 0.1495 | -4.6142 | -0.7913 | 0.2677 | 16.81 | 0.2293 | 0.1134 | 0.1581 | 0.5612 | 0.2728 |
| Att2Seq | 63.91 | 0.2099 | 0.2658 | 0.2379 | -4.5316 | -0.6707 | 0.7583 | 15.62 | 0.1583 | 0.1074 | 0.1147 | 0.5616 | 0.2470 |
| PETER | 67.00 | 0.2102 | 0.2983 | 0.2513 | -4.4100 | -0.5816 | 0.8750 | 15.57 | 0.3315 | 0.1298 | 0.2230 | 0.5800 | 0.3555 |
| PEPLER | 67.54 | 0.2920 | 0.3183 | 0.3052 | -4.4563 | -0.3354 | 0.9143 | 14.18 | 0.1476 | 0.1044 | 0.1050 | 0.5777 | 0.2524 |
| XRec | 74.53 | **0.3946** | 0.3506 | 0.3730 | -4.3911 | -0.2287 | **1.0000** | 11.45 | **0.0969** | 0.1048 | **0.0852** | 0.5770 | 0.2322 |
| G-Refer (7B) | 74.91 | 0.3573 | 0.4264 | 0.3922 | **-2.8716** | -0.1451 | **1.0000** | 10.88 | 0.1050 | **0.0952** | 0.0862 | 0.3571 | 0.2197 |
| G-Refer (8B) | **75.16** | 0.3629 | **0.4373** (+8.67%) | **0.4003** (+2.73%) | -2.8717 | **-0.1335** | **1.0000** | **10.76** | 0.1068 | 0.0995 | 0.0885 | **0.3534** | **0.2182** |
| **Google-reviews** | | | | | | | | | | | | | |
| NRT | 58.27 | 0.3509 | 0.3495 | 0.3496 | -4.2915 | -0.4838 | 0.2533 | 19.16 | 0.2176 | 0.1267 | 0.1571 | 0.6620 | 0.3118 |
| Att2Seq | 61.31 | 0.3619 | 0.3653 | 0.3636 | -4.2627 | -0.4671 | 0.5070 | 17.47 | 0.1855 | 0.1247 | 0.1403 | 0.6663 | 0.3198 |
| PETER | 65.16 | 0.3892 | 0.3905 | 0.3881 | -4.1527 | -0.3375 | 0.4757 | 17.00 | 0.2819 | 0.1356 | 0.2005 | 0.6701 | 0.3272 |
| PEPLER | 61.58 | 0.3373 | 0.3711 | 0.3546 | -4.1744 | -0.2892 | 0.8660 | 17.17 | 0.1134 | 0.1161 | 0.0999 | 0.6752 | 0.2484 |
| XRec | 69.12 | **0.4546** | 0.4069 | 0.4311 | -4.1647 | -0.2437 | 0.9993 | 14.24 | **0.0972** | 0.1163 | 0.0938 | 0.6591 | 0.2452 |
| G-Refer (7B) | 71.47 | 0.4253 | 0.4873 | 0.4566 | **-2.6013** | -0.1561 | **1.0000** | 13.46 | 0.1184 | **0.0872** | 0.0921 | 0.4004 | **0.2415** |
| G-Refer (8B) | **71.73** | 0.4245 | **0.4935** (+7.48%) | **0.4592** (+2.81%) | -2.6117 | **-0.1517** | **1.0000** | **13.23** | 0.1175 | 0.0920 | **0.0916** | **0.3921** | 0.2511 |



**Figure 3: Human evaluation comparing XRec and G-Refer.**

For retrieval-augmented fine-tuning, we use the models from the open-source LLaMA family. Specifically, for a fair comparison, we utilize the same model, i.e., LLaMA 2-7B adopted by baselines [39]. We also report results based on the advanced LLaMA 3-8B model. We set the learning rate, epochs, and max length as 2e-5, 2, and 2048 for RAFT, which is trained on 8 NVIDIA A100 GPUs. The total batch sizes are set to 32 and 16, respectively, for the 7B and 8B models. The rank of the LoRA adapter is set to 8. For inference, we set the temperature as 0 and the maximum output tokens as 256, ensuring a stable and reasonable generation. In addition, we employ the GPT-3.5-turbo model for computing the GPTScore metric.

## 4.2 Model Performance (RQ1)

*4.2.1 Overall Performance.* We first compare the quality of generated explanations against baselines across three datasets, the results are summarized in Table 2. It is observed that G-Refer demonstrates superior performance in both explainability and stability, outperforming baselines across semantic evaluators such as GPT, BERT, and BART. Compared to the most powerful baseline, XRec [39],

which uses GNNs to implicitly capture CF information, our model shows improvements in $BERT_{score}^{F1}$ across three datasets, with increases of 1.67%, 2.73%, and 2.81% respectively. This indicates that the explicit CF information retrieved by our hybrid graph retriever is more accurate and better utilizes both structures and semantics of user-item interaction graphs compared to the implicit CF information captured by XRec. In addition, graph translation enables the adapter-free RAFT, allowing LLMs to better comprehend the structural inputs and leverage their powerful capabilities to generate human-understandable explanations. Notably, G-Refer significantly outperforms all baselines in $BERT_{score}^R$, with increases of 4.56%, 8.67%, and 7.48% across the three datasets, while slightly decreasing in $BERT_{score}^P$. The retrieved CF information enables generated explanations to include more key information (e.g., user topic modeling and interaction history). Although introducing retrieved knowledge inevitably introduces some noise, reducing the precision slightly, we consider this trade-off beneficial, as the completeness of explanations is crucial for user understanding compared to accurate but less informative expressions. Finally, we notice that G-Refer's performance improvement on Amazon is relatively modest compared to other datasets. This can be attributed to the sparsity of its user-item interaction graph, particularly in the test set where the average node degree is only 2.76. In such cases, even with retrieval, the effectiveness of CF information from the graph is limited.

*4.2.2 Human Evaluation.* The ultimate goal of model explanation is to aid human understanding and decision-making. Human evaluation is thus the best way to evaluate the effectiveness of an explainer, which has been widely used in previous works [18, 48]. We conduct a human evaluation by randomly selecting 20 user-item

**Table 3: Ablation study for G-Refer, with the best results highlighted red and the worst in blue for the component.**

| Datasets | Yelp | | Google-reviews | |
|---|---|---|---|---|
| Ablations | $\text{BERT}^{\text{F1}} \uparrow$ | $\text{BERT}_{\text{std}} \downarrow$ | $\text{BERT}^{\text{F1}} \uparrow$ | $\text{BERT}_{\text{std}} \downarrow$ |
| *Variants on graph retriever:* | | | | |
| w/o path-level | 0.3927 | 0.0868 | 0.4560 | 0.0924 |
| w/o node-level | 0.3966 | 0.0894 | 0.4544 | 0.0922 |
| w/o GraphRAG | 0.3880 | 0.0896 | 0.4468 | 0.0940 |
| *Variants on link prediction model:* | | | | |
| w/ LightGCN | 0.3941 | 0.0870 | 0.4589 | 0.0922 |
| w/ R-GCN | 0.4003 | 0.0885 | 0.4592 | 0.0916 |
| *Variants on LLMs with different scales:* | | | | |
| w/ Qwen-0.5B | 0.3201 | 0.6530 | 0.4129 | 0.2171 |
| w/ Qwen-1.5B | 0.3557 | 0.3940 | 0.4451 | 0.0940 |
| w/ Qwen-3B | 0.3994 | 0.0861 | 0.4602 | 0.0903 |
| w/ Qwen-7B | 0.3991 | 0.0851 | 0.4582 | 0.0914 |
| *Knowledge pruning v.s. full training set:* | | | | |
| w/o pruning | 0.4002 | 0.0892 | 0.4605 | 0.0909 |
| G-Refer | 0.4003 | 0.0885 | 0.4592 | 0.0916 |

pairs from the test set of each dataset and generating explanations for each sample using XRec [39] and G-Refer. We designed a survey with single-choice questions and distributed it to five senior researchers, asking them to select the best explanations. As shown in Figure 3, explanations generated by G-Refer were consistently favored across all datasets, especially for Yelp and Google-reviews, where our explanations were chosen in over 80% of cases.

*4.2.3 Case Study.* To demonstrate the effectiveness of G-Refer and show how retrieved CF signals benefit the generated explanations, we provide several cases and give analysis in Appendix B.

### 4.3 Ablation Study (RQ2)

To investigate how each component affects the model performance, we conduct the following ablation studies in Table 3.

*4.3.1 The Effects of Hybrid Graph Retriever.* From Table 3 we can observe that both path-level and node-level retrievers contribute to the final results, with their combination yielding the best performance. We observe that semantic information is more crucial for Yelp, while structural knowledge offers greater advantages for Google-reviews. This suggests that the importance of retrievers may differ based on dataset characteristics and highlights the complementarity of the two retrievers.

*4.3.2 The Effects of Various GNNs.* The GNN encoder in our path-retriever captures structural CF information and can be replaced with any advanced GNN-based recommender model. We compare LightGCN [20], which is one of the representative GNN models for recommendation, with R-GCN [49] in the ablation study. The results show that both of the GNNs achieve competitive performance. This demonstrates their effectiveness in capturing structural information for recommendation and the flexibility of our path-level retriever, which could adapt to different graphs and recommendation scenarios using different GNN architectures.

*4.3.3 The Effects of Various LLMs with Different Scales.* We investigate the impact of LLM scales on G-Refer's performance by

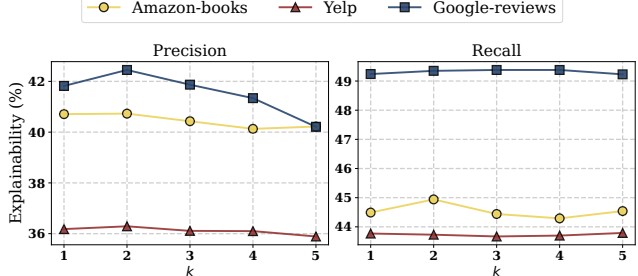

**Figure 4: Performance of different retrieved number $k$.**

comparing the advanced Qwen 2.5 family [61] of different scales. Firstly, we observe a clear performance scaling from 0.5B to 3B models. Interestingly, the 3B model achieves comparable results to the 7B model, indicating that with the introduction of retrieved knowledge, even a relatively smaller LLM can achieve impressive results through RAFT. This also demonstrates the necessity of fine-tuning with retrieved knowledge, which can significantly enhance the model's ability to generate high-quality explanations.

*4.3.4 The Effects of Knowledge Pruning.* Training with the full dataset increases the number of samples by several times but does not improve performance proportionally, and may even lead to slight degradation (e.g., on Yelp). This is because some user-items are self-explanatory, and additional CF information may introduce noise. Thus, knowledge pruning is essential for removing these samples from training and improving model performance.

### 4.4 Hyperparameter Study (RQ3)

We explore the variation of G-Refer's performance with respect to two hyperparameters: $k$ (the number of retrieved elements, including both paths and nodes) and $t$ (the knowledge pruning ratio).

*4.4.1 Retrieved Number $k$.* Figure 4 illustrates the impact of $k$ (ranging from 1 to 5) on $\text{BERT}_{\text{score}}$ precision and $\text{BERT}_{\text{score}}$ recall (abbr. precision and recall in the following). The best precision is achieved at $k$=2, after which it decreases, showing that more retrieved paths and nodes can introduce noise and deteriorate the performance. Conversely, a low $k$ (i.e., $k$=1) fails to provide sufficient CF information, leading to lower results. We also notice that recall is not sensitive to the increased $k$, suggesting that a modest $k$ can effectively retrieve essential information for generating explanations.

*4.4.2 Pruning Ratio $t$.* In Table 4, we report G-Refer's explainability and stability on three datasets with $t$ ranging from 90% to 50%. In terms of explainability, both Amazon-books and Yelp perform best at $t = 70\%$, while Google-reviews at $t = 50\%$. This suggests that a low $t$ value, which corresponds to retaining more training samples, does not always lead to improved model performance, as irrelevant samples can introduce noise. However, it generally enhances model stability, indicating that fine-tuning with more data can enhance the model's ability to generate consistent explanations.

### 4.5 Efficiency Analysis (RQ4)

In Figure 5, we analyze the efficiency of our model, including the time taken to train one epoch, performance results, and the number

**Table 4: Performance of different knowledge pruning ratio $t$.**

| Ratio | Explainability ↑ | | | Stability ↓ | | |
|---|---|---|---|---|---|---|
| | BERT$^{F1}$ | BART | BLEURT | BERT$_{std}$ | BART$_{std}$ | BLEURT$_{std}$ |
| **Amazon-books** | | | | | | |
| $t$=90% | 0.4211 | -2.5045 | -0.1388 | **0.0817** | 0.2811 | 0.1905 |
| $t$=80% | 0.4253 | -2.5051 | -0.1339 | 0.0821 | 0.2803 | 0.1913 |
| $t$=70% | **0.4289** | 2.4799 | **-0.1203** | 0.0825 | 0.2791 | 0.1912 |
| $t$=60% | 0.4279 | **-2.4743** | 0.1263 | 0.0830 | 0.2799 | 0.1910 |
| $t$=50% | 0.4255 | -2.4900 | -0.1326 | 0.0819 | **0.2744** | **0.1893** |
| **Yelp** | | | | | | |
| $t$=90% | 0.3974 | -2.9125 | -0.1455 | 0.0872 | 0.3724 | 0.2240 |
| $t$=80% | 0.3961 | -2.8893 | -0.1458 | 0.0866 | 0.3591 | 0.2207 |
| $t$=70% | **0.4003** | **-2.8717** | **-0.1335** | 0.0885 | 0.3534 | 0.2182 |
| $t$=60% | 0.3927 | -2.8859 | -0.1461 | **0.0856** | 0.3559 | 0.2178 |
| $t$=50% | 0.3967 | -2.8934 | -0.1461 | **0.0856** | **0.3518** | **0.2164** |
| **Google-reviews** | | | | | | |
| $t$=90% | 0.4462 | -2.6987 | -0.1798 | 0.0919 | 0.4042 | 0.2437 |
| $t$=80% | 0.4518 | -2.6202 | -0.1645 | 0.0927 | 0.3912 | 0.2508 |
| $t$=70% | 0.4592 | 2.6117 | -0.1517 | 0.0916 | 0.3921 | 0.2511 |
| $t$=60% | 0.4583 | -2.6169 | -0.1465 | 0.0918 | 0.3908 | **0.2477** |
| $t$=50% | **0.4606** | **-2.6073** | **-0.1459** | **0.0907** | **0.3852** | 0.2498 |

of parameters that require tuning for G-Refer, XRec [39], and full-set training. For a fair comparison, all experiments were conducted on a single GPU with a batch size of 1. Though requiring more learnable parameters, G-Refer achieves faster training and better performance compared to XRec. This can be attributed to (1) our knowledge pruning which significantly reduces training data; (2) XRec's speed limitation due to adapter insertion. In addition, full-set training reduces efficiency without performance gains, indicating the necessity of knowledge pruning.

# 5 RELATED WORKS

## 5.1 Explainable Recommendation

Explainable recommendation [45, 70, 71] has demonstrated significant advantages in informing users about the logic behind recommendations, thereby increasing system transparency, effectiveness, and trustworthiness. Early works focus on generating explanations with predefined templates [29] or extracting logic reasoning rules from recommendation models [5, 51, 75]. To provide more detailed and personalized explanations, recent works have explored generating explanations from graph structure, which contains abundant collaborative information (CF) for providing explanations [1, 15, 59]. For example, PGPR [59] proposes a reinforcement learning-based method to find a path in the graph to explain the recommendation. However, the graph structure is often complex and hard for users to understand. To address this issue, some works have explored generating explanations in natural language [4, 8, 9, 11, 14, 30–32, 32, 39]. With the advance of large language models (LLMs), like ChatGPT [43], they can generate more fluent and informative explanations based on user's and item's profile [31]. Recently, researchers have tried to combine the advantages of graphs to enhance the explanation generated by LLMs. XRec [39] adopts a graph neural network (GNN) to model the graph structure and generate embeddings. Then, the embeddings are fed into LLMs to generate explanations. This approach allows LLMs to produce more informative explanations by considering CF information within the graph structure. However, XRec represents CF information as hidden embeddings,

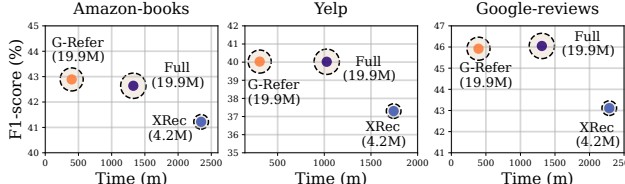

**Figure 5: Efficiency Analysis of G-Refer.**

leaving it unclear what specific CF information is considered when generating explanations. This ambiguity makes it difficult to verify the explanations generated by LLMs, which suffer from severe hallucination issues [72] and further diminishes the trustworthiness of the explainable recommendation.

## 5.2 Graph Retrieval-Augmented Generation

Although retrieval-augmented generation (RAG) has been widely used to enhance the LLMs by incorporating external knowledge [17], it has shown limitations in considering the graph structure. To address this issue, graph retrieval-augmented generation (GraphRAG) has been proposed to effectively incorporate graph information into the LLM generation process [46]. The GraphRAG typically involves two processes: *graph retrieval* and *graph-augmented generation*. In graph retrieval, we need to design a retrieval model to retrieve relevant graph information for the given query, which can be some simple non-parametric retrievers like $L$-hop neighbors. To better consider the graph structure, GNN-based retrievers have been proposed. For example, GNN-RAG [40] first encodes the graph, assigns a score to each entity, and retrieves entities relevant to the query. Instead of GNNs, RoG [38] proposes an LLM-based graph retriever by planning, retrieving, and reasoning on graphs. In the graph-augmented generation, the retrieved graph information is used to enhance the generation process of LLMs. Due to the unstructured nature of the LLMs, some works introduce an adapter layer to bridge the modality gap. GraphLLM [3] first encodes the graph as embeddings and then feeds them into LLMs with a graph adapter layer. However, the hidden embeddings used by these methods are difficult to interpret and understand. To improve the interpretability, recent research adopts some graph description language to translate the graph structure into natural language, such as edge list, node list, and syntax tree [25, 73]. In this way, the graph structure can be understood by both LLMs and humans. Thus, we adopt a simple prompt to describe the graph structure for the explainable recommendation.

# 6 CONCLUSION

We systematically analyze the limitations of existing explainable recommendation methods and introduce G-Refer, a novel model to address these challenges. The key design of our model involves leveraging a hybrid graph retrieval to extract explicit CF signals, employing knowledge pruning to filter out less relevant samples, and utilizing retrieval-augmented fine-tuning to integrate retrieved knowledge into explanation generation. Comprehensive experiments validate the effectiveness of our model. G-Refer reveals great potential for graph retrieval-augmented generation in recommendation scenarios. Future work includes exploring fully training-free retrievers and investigating the transferability to other tasks.

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

# A MORE EXPERIMENT DETAILS

## A.1 Details of Datasets

- **Amazon-books** [41] is a subset of the Amazon review dataset, specifically focused on book recommendations. This dataset encompasses user-book interactions, including both numerical ratings and textual reviews submitted by users after their purchases.
- **Yelp** [39] is a widely used dataset derived from Yelp[1], where local businesses such as restaurants and bars are considered items. It provides rich information about local businesses, including user reviews and ratings across various categories.
- **Google-reviews** [28, 60] comprises restaurant reviews collected from Google Local[2]. This dataset incorporates both business metadata and user feedback, offering a broad perspective on dining establishments globally.

## A.2 Details of Metrics

- **GPT$_{\text{score}}$** [54] leverages large language models to evaluate text quality, providing a context-aware assessment.
- **BERT$_{\text{score}}$** [68] computes the similarity between reference and generated texts using contextual embeddings from BERT [12]. Given a reference sentence $x = \langle x_1, x_2, ..., x_n \rangle$ and a generated sentence $\hat{x} = \langle \hat{x}_1, \hat{x}_2, ..., \hat{x}_m \rangle$, A sequence of word embeddings are first generated using BERT:

$$\begin{aligned} \text{BERT}(\langle x_1, x_2, ..., x_n \rangle) &= \langle \mathbf{x_1}, \mathbf{x_2}, ..., \mathbf{x_n} \rangle \\ \text{BERT}(\langle \hat{x}_1, \hat{x}_2, ..., \hat{x}_m \rangle) &= \langle \mathbf{\hat{x}_1}, \mathbf{\hat{x}_2}, ..., \mathbf{\hat{x}_m} \rangle \end{aligned} \tag{13}$$

The similarity between two individual embeddings $(\mathbf{x_i}, \mathbf{\hat{x}_j})$ is measured using cosine similarity, which simply reduces to $\mathbf{x_i}^\top \mathbf{\hat{x}_j}$ since both embeddings are pre-normalized. With these definitions, the Precision, Recall, and F1-score are calculated as follows:

$$\text{BERT}_{\text{score}}^{\text{P}} = \frac{1}{|\hat{x}|} \sum_{\hat{\mathbf{x}}_\mathbf{j} \in \hat{x}} \max_{\mathbf{x_i} \in x} \overbrace{\mathbf{x_i}^\top \mathbf{\hat{x}_j}}^{\text{cosine similarity}} \tag{14}$$

$$\text{greedy matching}$$

$$\text{BERT}_{\text{score}}^{\text{R}} = \frac{1}{|x|} \sum_{\mathbf{x_i} \in x} \max_{\hat{\mathbf{x}}_\mathbf{j} \in \hat{x}} \overbrace{\mathbf{x_i}^\top \mathbf{\hat{x}_j}}^{\text{cosine similarity}} \tag{15}$$

$$\text{greedy matching}$$

$$\text{BERT}_{\text{score}}^{\text{F1}} = 2 \times \frac{\text{BERT}_{\text{score}}^{\text{P}} \times \text{BERT}_{\text{score}}^{\text{R}}}{\text{BERT}_{\text{score}}^{\text{P}} + \text{BERT}_{\text{score}}^{\text{R}}} \tag{16}$$

- **BART$_{\text{score}}$** [66] conceptualizes the evaluation as a text generation task, assigning scores based on the probability of regenerating reference texts using the BART model [26].
- **BLEURT** [50] employs a novel language model pre-trained with synthetic data to assess the similarity between the generated and reference texts.
- **USR** [30] assesses the uniqueness of generated explanations by calculating the ratio of unique sentences to total sentences.

---

[1]https://www.yelp.com/dataset/
[2]https://www.google.com/maps

## A.3 Details of Baselines

- **NRT** [32] employs multi-task learning to predict ratings and generate tips for recommendations simultaneously based on user and item IDs. The generation component is a GRU [10].
- **Att2Seq** [14] implements an attention-based attribute-to-sequence model that generates reviews by leveraging attribute information. The generation component is a two-layer LSTM [19].
- **PETER** [30] is a personalized Transformer model that maps user and item IDs to generated explanations. It bridges IDs and words through a "context prediction" task. PETER is used instead of PETER+ due to the absence of word features in the datasets.
- **PEPLER** [31] proposes sequential tuning and recommendation as regularization strategies to bridge the gap between prompts (incorporating user and item ID vectors) and the pre-trained transformer model for generating explanations.
- **XRec** [39] utilizes the encoded user/item embeddings from GNNs as implicit collaborative signals, which are then integrated into each layer of LLMs, enabling the generation of explanations.

# B CASE STUDY

We present two cases in Table 5 and Table 6 to demonstrate the effectiveness of G-Refer and illustrate how retrieved CF signals benefit the generated explanations. All retrieved knowledge is presented as human-readable text after graph translation for better user understanding. We also provide the ground truth explanations and explanations generated by XRec [39] for comparison.

From Table 5, we can observe that the user profile contains no explicit drinking interests. However, node-level CF signals reveal that other users who have similar interests to the current user show a preference for drinking. This allows G-Refer to infer that the current user might appreciate this restaurant specializing in "crafted beers". In contrast, XRec's explanation is notably generic, which relies on common phrases like "delicious food" and "great service", missing the key recommendation reasons, thus leading to a low Recall score. Table 6 provides another case that leverages path-level retrieved CF information for more comprehensive explanations. While the user profile only indicates general preferences for comfort food and efficient services, path-level CF signals uncover broader connections between the current user and other users/items. Users along the retrieved path have an interest in "frozen yogurt", allowing us to infer that the current user might be interested in this restaurant due to its frozen yogurt offerings. In contrast, XRec incorrectly explains that the user might choose this restaurant for its Thai food, failing to capture the user's true interests. These cases show that node-level and path-level CF signals are complementary. While not equally effective in every situation, gathered CF signals can contribute to generating more accurate explanations in recommendation scenarios. The better explanations demonstrate G-Refer has the ability to effectively retrieve, understand, and utilize such knowledge.

**Table 5: A case from Yelp, which shows how G-Refer primarily leverages node-level retrieved CF information to generate better explanations.**

| | Inputs |
|---|---|
| **User Profile** | The user is likely to enjoy Mexican restaurants with a focus on burritos and tacos, as well as baseball games at scenic locations and coffee shops for cozy work or socializing. |
| **Item Profile** | Beer enthusiasts looking for a laid-back atmosphere with a variety of craft beers and casual dining options, including dog owners seeking a pet-friendly environment, would enjoy this business. |
| **Node-level CF** | For the user-item pair, here are some related users and items: Users: The user is likely to enjoy businesses with unique decor, traditional family recipes, craft beers, outdoor dining, gourmet wood-fired entrees, live music, and waterfront locations., The user is likely to enjoy fine dining experiences, laid-back accommodations with resort amenities, American pub fare with craft beer, and relaxed atmospheres with delicious comfort food and craft beer selections. Items: Coffee enthusiasts looking for a cozy atmosphere with delicious pastries and brunch options would enjoy Lulu Carpenter's., Food enthusiasts who appreciate classic Italian cuisine and casual dining experiences would enjoy Joe's Pizza & Subs. |
| | **Explanations** |
| **Ground Truth** | The user would enjoy the business for its awesome selection of beers and ciders, great outdoor spaces, and the option to order bar food from Carpos, even though it predominantly focuses on beer. |
| **XRec** | The user would enjoy the business because of its delicious food, great service, and cozy atmosphere, making it a great place to relax and enjoy a good meal. |
| **G-Refer (ours)** | The user would enjoy the business because of the great selection of beers, friendly staff, and the opportunity to try new and unique brews like the hazy IPA. |

**Table 6: A case from Google-review, which shows how G-Refer primarily leverages path-level retrieved CF information to generate better explanations.**

| | Inputs |
|---|---|
| **User Profile** | This user is likely to enjoy businesses that offer comfort food with unique twists, efficient waxing services with minimal pain, upscale dining with stunning views and impressive service, and flavorful Asian cuisine with fast service in Philadelphia. |
| **Item Profile** | Fans of frozen yogurt with a wide variety of flavors and toppings, in a vibrant and welcoming setting in Philadelphia, would enjoy Berry Sweet Frozen Yogurt. Ideal for those who appreciate a fun atmosphere, quality treats, and convenient location on South Street. |
| **Path-level CF** | For the given user-item pair, here are several related paths connecting users and items through their interactions: 1. User (Profile: This user is likely to enjoy businesses that offer comfort food with unique twists, efficient waxing services with minimal pain, upscale dining with stunning views and impressive service, and flavorful Asian cuisine with fast service in Philadelphia.) -> buys -> Item (Profile: Users who appreciate high-quality skincare products, attentive customer service, and are willing to splurge for effective beauty items would enjoy Kiehl's in Philadelphia.) -> bought by -> User (Profile: This user is likely to enjoy businesses offering high-quality food and attentive service in cozy settings, such as restaurants like Kanella, Village Whiskey, and Cheu Noodle Bar. They appreciate flavorful meals, knowledgeable staff, and unique dining experiences.) -> buys -> Item (Profile: Users who appreciate creative and high-quality vegan options, a personalized dining experience with small restaurant owners, unique and experimental dishes with varied influences, and a great wine and cocktail selection would enjoy Meritage in Philadelphia.) -> bought by -> User (Profile: The user is likely to enjoy upscale dining with Latin American and Mexican cuisine, diverse frozen yogurt flavors with unique bubble tea options, good food in a low-key ambiance, and authentic spicy Indian cuisine despite some reservations about service and chicken quality.) -> buys -> Item (Profile: Fans of frozen yogurt with a wide variety of flavors and toppings, in a vibrant and welcoming setting in Philadelphia, would enjoy Berry Sweet Frozen Yogurt. Ideal for those who appreciate a fun atmosphere, quality treats, and convenient location on South Street.) |
| | **Explanations** |
| **Ground Truth** | The user would enjoy Berry Sweet because of its delicious frozen yogurt, friendly staff, variety of toppings, and fun atmosphere that makes it a great place to hang out with friends on South Street. |
| **XRec** | You would enjoy the business because it offers delicious, authentic Thai food with a variety of dishes to choose from, including spicy and non-spicy options, as well as a cozy atmosphere with friendly service. |
| **G-Refer (ours)** | The user would enjoy Berry Sweet because of the wide variety of flavors and the opportunity to mix and match them, making it a fun for yogurt lovers seeking a customizable and fun experience. |

