# OpenReview forum: "G-Refer: Graph Retrieval-Augmented Large Language Model for Explainable Recommendation"
_ACM.org/TheWebConf/2025/Conference — WWW 2025 Oral_

### Official Review · Reviewer_aYvV · 2024-11-19

**Novelty:** 5
**Technical Quality:** 6

**Review:**

This work introduces G-Refer, an LLM provides explanation for a user-item pair given their profiles and user-item interaction graph. Prior work, e.g., XRec, uses implicit CF signals and does not capture well the explanation information from the given context. G-Refer is provided with more explicit information for generating explanation, which is extracted by a hybrid graph retrieval mechanism and further condensed using a knowledge pruning approach. The LLM is fine-tuned using retrieval-augmented fine-tuning to ensure desirable explanation output. Experiments show case the effectiveness of the proposed framework in terms of generating better explanation texts (using GPT_score, BERT_score, BART_score, BLEURT, and USR). The ablation study is sufficiently show the contributions of proposed components as well as the comparisons among variants.

**Pros**
- (P1) The problem is well motivated and systematically addressed using various components.
- (P2) The source code for reproducibility is provided.
- (P3) Experiments show superior performance of the proposed method over the baseline method.

**Cons**
- (C1) This work try to produce better explanation for a given recommendation. It does not try to enhance recommendation accuracy. While "explainable recommendation" model used (in many literature) to describe a model that generate not only recommendations but also explanations, I think the wording usage for this work should be "recommendation explanation" since the proposed model does not try to enhance the recommendation accuracy.
- (C2) The improvement in some metrics is marginal (e.g., BERT_F1_score). The improvement % in Table 2 is not well defined. It can only be found in the paragraph, which is the different between the performance of G-Refer and XRec.
- (C3) Except USR metric (unique sentence ratio), other utilized metrics (GPT_score, BERT_score, BART_score, BLEURT) evaluate the natural text generation quality based on learned language model. Which makes the interpretation of the quality of generated text difficult to explain. Although the human evaluation bridges this gap, other evaluation measurements related to coherence, conciseness, and diversity can be considered for evaluation.

**Questions:**

- (Q1) Does the statistical test show the outperformance over baselines (for metrics such as GPT_score, BERT_F1_score, BART_score, BLEURT)?

- (Q2) Why can G-Refer produce better explanation for recommendation?

- (Q3) What are the limitation of the proposed approach (G-Refer)?

**Reviewer Confidence:**

4: The reviewer is certain that the evaluation is correct and very familiar with the relevant literature

**Scope:**

4: The work is relevant to the Web and to the track, and is of broad interest to the community

---

### Official Review · Reviewer_2aue · 2024-11-20

**Novelty:** 4
**Technical Quality:** 4

**Review:**

This paper proposes the G-Refer framework, which combines a hybrid graph retrieval mechanism with lightweight retrieval-enhanced fine-tuning to help large language models better understand complex collaborative filtering information in graphs, thereby generating more accurate, context-aware, and personalized recommendation explanations. Experimental results validate the effectiveness of G-Refer, demonstrating its significant advantages in improving the quality of recommendation explanations.
Strengths:
[1]	G-Refer proposes a hybrid graph retrieval mechanism combining multi-granularity retrieval to extract structured and semantically rich collaborative filtering (CF) information from user-item interaction graphs.
[2]	By leveraging RAFT (Retrieval-Enhanced Fine-Tuning), it guides large language models to understand the CF information retrieved from graphs and generate more accurate recommendation explanations.

Weaknesses:
[1]	As far as I know, the test set used by XRec for calculating BERTScore does not include understanding of higher-order user-item interactions. Whether introducing this knowledge during LoRA fine-tuning, as your main innovation, brings more effective explanations is debatable.
[2]	XRec does not provide evaluation code for BART and BLEURT, while G-Refer shows such significant improvements in these two metrics. When I tested XRec using your method, the results outperformed the existing ones. Please explain this phenomenon and re-examine whether you have compared XRec fairly.
[3]	XRec incorporates collaborative signals, but you have not highlighted the differences between your approach to this problem and existing methods. Moreover, traces of XRec are pervasive in this paper. Please appropriately compare your work with the baselines.

If you can address my concerns effectively, I will consider increasing the score.

**Questions:**

Why does your approach work well? Please provide a proper theoretical analysis.

**Reviewer Confidence:**

4: The reviewer is certain that the evaluation is correct and very familiar with the relevant literature

**Scope:**

4: The work is relevant to the Web and to the track, and is of broad interest to the community

---

### Official Review · Reviewer_nb9W · 2024-11-29

**Novelty:** 5
**Technical Quality:** 5

**Review:**

Summary:
This paper introduces G-Refer, a framework leveraging graph retrieval-augmented large language models (LLMs) to improve explainable recommendation systems. The framework integrates hybrid graph retrieval, graph translation, and retrieval-augmented fine-tuning to address challenges related to graph complexity and the modality gap. Experiments demonstrate G-Refer’s superiority in explainability and stability across multiple datasets.
Strengths:
1.High Innovation: G-Refer employs a hybrid graph retrieval mechanism to explicitly extract collaborative filtering (CF) signals from structural and semantic perspectives. This approach overcomes the limitations of implicit node embeddings, making the recommendations more interpretable and transparent to users.
2.Comprehensive Validation: The paper provides extensive experimental results on multiple public datasets (e.g., Amazon-books, Yelp, Google-reviews), with metrics covering explainability and stability. The results highlight significant improvements in semantic-based evaluation metrics like BERTScore and GPTScore, showing the robustness of the framework.
3.Well-Designed Framework: The framework incorporates knowledge pruning (to reduce redundant samples) and retrieval-augmented fine-tuning (to improve the model’s ability to leverage retrieved information). These design choices enhance training efficiency while ensuring the quality and accuracy of the generated explanations, making the framework both effective and computationally efficient.

**Questions:**

1. There are some grammar errors: Some technical terms, such as "hybrid graph retrieval" and "knowledge pruning," are not adequately defined upon first mention, which may hinder understanding for non-experts. Additionally, certain sentences are overly long and dense, making the text difficult to follow. Simplifying expressions and improving logical transitions would enhance readability.
2. Insufficient Motivation for the Research Problem: While the paper addresses the trending topic of explainable recommendations, it lacks a thorough explanation of how "hybrid graph retrieval" and "graph translation" directly address gaps in existing methods. In particular, the uniqueness and advantages of the proposed method over related works like XRec need further clarification.
3. Methodology Not Fully Aligned with Motivation: The paper suggests that "hybrid graph retrieval" addresses the modality gap, but it does not sufficiently explain how structural and semantic information work together to achieve this. The alignment between the framework’s design and its theoretical goals could be more explicitly articulated to avoid any inconsistencies.
4. Incomplete Experimental Validation: The paper lacks detailed validation of the independent contributions of each module, such as the isolated impact of knowledge pruning and graph translation. While G-Refer outperforms baseline models (e.g., XRec), it does not provide direct comparisons under consistent settings to clarify the source of its advantages. Moreover, critical components like hybrid graph retrieval and knowledge pruning are not systematically analyzed in terms of their configurations (e.g., retrieval strategies and pruning ratios), leaving the full potential of these designs unexplored.

**Reviewer Confidence:**

3: The reviewer is confident but not certain that the evaluation is correct

**Scope:**

4: The work is relevant to the Web and to the track, and is of broad interest to the community

---

### Official Review · Reviewer_uTD5 · 2024-12-02

**Novelty:** 5
**Technical Quality:** 5

**Review:**

The paper introduces G-Refer, a framework that combines graph retrieval mechanisms with large language models for explainable recommendation systems. G-Refer addresses challenges in extracting explicit and semantically rich collaborative filtering information from user-item interaction graphs, enhancing the accuracy and personalization of recommendation explanations. Key contributions include a comprehensive analysis of existing limitations, the architecture design of G-Refer for capturing structural and semantic signals, and superior performance demonstrated through experiments.

Concerns:
1. The model's utilization of user profiles and item attributes remains unclear in the model description. Regarding explanation path retrieval, the method for computing weights across different edges and defining edge types needs clarification.Besides, how do you define the type of edges?
2. While the authors claim superiority over state-of-the-art baselines, a more competitive comparison could involve GCN models  that incorporate item information, such as like RGCN or A2GCN (Attribute-aware attentive GCN model, TKDE 2020) or more recent ones. Otherwise, it should adopt a more rigorous expression, like SOTA LLM-based explanation recommendation models

**Questions:**

Please ansser the first concern.

**Reviewer Confidence:**

4: The reviewer is certain that the evaluation is correct and very familiar with the relevant literature

**Scope:**

4: The work is relevant to the Web and to the track, and is of broad interest to the community